# Why does the number of antenatal care visits in Ethiopia remain low?: A Bayesian multilevel approach

Daniel Atlaw[1]*, Tesfaye Getachew Charkos[2], Jeylan Kasim[1], Vijay Kumar Chatu[3,4]

1 Public Health Department, School of Health Science, Madda Walabu University, Bale Goba, Ethiopia, 2 School of Public Health, Adama Hospital Medical College, Adama, Ethiopia, 3 Center for Transdisciplinary Research, Saveetha Dental College, Saveetha Institute of Medical and Technical Sciences (SIMATS), Saveetha University, Chennai, India, 4 Center for Evidence-Based Research, Global Health Research and Innovations Canada (GHRIC), Toronto, ON, Canada

* danielatmwu@gmail.com

**Data Availability Statement:** The data underlying the results presented in the study are available from (https://www.dhsprogram.com/Data/).

**Funding:** The author(s) received no specific funding for this work.

## Abstract

### Introduction

Antenatal care (ANC) visit is a proxy for maternal and neonatal health. The ANC is a key indicator of access and utilization of health care for pregnant women. Recently, eight times ANC visits have been recommended during the pregnancy period. However, nearly 57% of women received less than four ANC visits in Ethiopia. Therefore, the objective of this study is to identify factors associated withthe number of ANC visits in Ethiopia.

### Methods

A community-based cross-sectional study design was conducted from March 21 to June 28/ 2019. Data were collected using interviewer-administered questionnaires from reproductive age groups. A stratified cluster sampling was used to select enumeration areas, households, and women from selected households. A Bayesian multilevel negative binomial model was applied for the analysis of this study. There is an intra-class correlation (ICC) = 23.42% and 25.51% for the null and final model, respectively. Data were analyzed using the STATA version 17.0. The adjusted incidence risk ratio (IRR) with 95% credible intervals (CrI) was used to declare the association.

### Result

A total of 3915 pregnant women were included in this study. The mean(SD) age of the participants was 28.7 (.11) years. Nearly one-fourth (26.5%) of pregnant women did not have ANC visits, and 3% had eight-time ANC visits in Ethiopia. In the adjusted model, the age of the women 25–28 years (IRR:1.13; 95% CrI: 1.11, 1.16), 29–33 years (IRR: 1.15; 95% CrI: 1.15, 1.16), ≥34 years (IRR:1.14; 95% CrI: 1.12, 1.17), being a primary school (IRR: 1.22, 95% CrI: 1.21, 1.22), secondary school and above (IRR: 1.26, 95% CrI: 1.26, 1.26), delivered in health facility (IRR: 1.93; 95% CrI: 1.92, 1.93), delivered with cesarian section (IRR: 1.18; 95% CrI: 1.18, 1.19), multiple (twin) pregnancy (IRR: 1.11; 95% CrI: 1.10, 1.12),

**Competing interests:** The authors have declared that no competing interests exist.

richest (IRR:1.23; 95% CrI: 1.23, 1.24), rich family (IRR: 1.34, 95% CrI: 1.30, 1.37), middle income (IRR: 1.29, 95% CrI: 1.28, 1.31), and poor family (IRR = 1.28, 95% CrI:1.28, 1.29) were shown to have significant association with higher number of ANC vists, while, households with total family size of $\geq 5$ (IRR: 0.92; 95% CrI: 0.91, 0.92), and being a rural resident (IRR: 0.92, 95% CrI: 0.92, 0.94) were shown to have a significant association with the lower number of ANC visits.

## Conclusion

Overall, 26.5% of pregnant women do not have ANC visits during their pregnancy, and 3% of women have eight-time ANC visits. This result is much lower as compared to WHO's recommendation, which states that all pregnant women should have at least eight ANC visits. In this study, the ages of the women 25–28, 29–33, and $\geq$34 years, being a primary school, secondary school, and above, delivered in a health facility, delivered with caesarian section, multiple pregnancies, rich, middle and poor wealth index, were significantly associated with the higher number of ANC visits, while households with large family size and rural residence were significantly associated with a lower number of ANC visits in Ethiopia.

## Introduction

Antenatal care (ANC) visit is a proxy for maternal and neonatal health. The ANC visit is a key indicator of access to and utilization of health care for pregnant women. It is one of the important components of maternal and child health care services for reducing maternal and neonatal mortality rates [1, 2]. Early initiation and encouraging regular visits are important to reduce pregnancy complications [3]. For instance, a systematic review in Ethiopia revealed that a focused quality ANC significately reduces neonatal mortality by 34% [4]. The main objective of ANC is the early identification of preexisting diseases or risk factors that occur during pregnancy and childbirth, as well as promoting the well-being of mothers and their newborns [4–6]. The ANC package is important to prepare for birth and avoid threats affecting mothers and babies during pregnancy [7].

About 15% of pregnancies result in life-threatening complications that require intervention by a skilled healthcare providers [8]. In middle and low-income countries, complications during pregnancy and childbirth are major causes of death and disability among reproductive-age women [9, 10]. To decrease the avoidable causes of maternal and neonatal mortality, WHO recomemded a minimum of eight or more ANC visits [11]. However, in Africa, women lack access to health facilities and quality ANC service due to distance, cultural beliefs, cost of transportation, and barriers in attitudes [12]. The utilization of eight or more ANC visits is only 6.8% in Sub-Saharan Africa [13]. About 32% of pregnant women did not receive ANC visits throughout their pregnancy period in Ethiopia [11]. Previous studies revealed that as the number of ANC visits increases, maternal and neonatal morbidity [14] and mortality were shown to decrease [15–18]. In Ethiopia, although ANC service coverage was increased, the recommended number of ANC visits was not achieved at the national level [15].

Currently, in Ethiopia, the overall coverage of ANC visits is nearly 74%, but the number of four ANC visits is around 43% [19]. In terms of access to health facilities, the ratio of health facilities including health post to pregnant women is 1:185, while the number of midwives to pregnant women is 1:162 [20]. Some studies identified factors like lack of information on the

importance of ANC, distance from health facilities, and culture as main contributors to low coverage [15, 21]. To increase the number of ANC visits, it is important to identify factors that affect ANC visits in Ethiopia.

Different studies were conducted in Ethiopia, but the previous studies were conducted on utilization level [22, 23], time of initiation [3, 24–27], and associated factors [3, 12, 26] rather than focusing on the number of visits, which may result in loss of information in classification count data. Further, the previous studies were conducted with the older recommendation of WHO, which emphasis on focused four ANC visits. Currently, WHO has revised the previous guideline to increase the number of ANC visits to eight or more [28]. Therefore, this study, aimed to identify community and individual-level determinants of the number of ANC visits in Ethiopia.

## Methods

### Study setting and design

Ethiopia is one of the East African countries, that has 10 regional states (Afar, Amhara, Benishangul-Gumuz, Gambella, Oromia, Somali, Southern Nations, Nationalities and People's (SNNP), Harari, Sidama, and Tigray), and two metropolis towns (Addis Ababa and Dire Dawa). It is estimated that the total population of Ethiopia is 105,163,988 [29]. Of this, 52,439,998 (4.98%) are women of reproductive age [29]. The Ethiopian Ministry of Health provides maternal health services free of charge. A cross-sectional community-based study design was conducted from March 21 to June 28 /2019, among reproductive-age women in Ethiopia.

### Data source and sampling

The Ethiopian Demographic and Health Survey (EDHS) data are collected nationally every 5 years on key indicators. Data were collected using an interviewer-administered questionnaire from reproductive age (15–49 years) women. The questionnaire includes socio-demographic, socioeconomic, and service-related maternal health variables [19]. The source of data for this analysis was from the DHS website (www.dhsprogram.com) after requesting permission by submitting a protocol for the study.

A two-stage stratified cluster sampling was employed to select a total of 305 enumeration areas (EAs) using probability proportional to EA size. Out of which 93 were urban and 212 were rural areas. Independent selection was applied in each sampling stratum [19].

Then a fixed number of 30 households per cluster were selected using systematic sampling. From each selected household all reproductive-age women were eligible for interview. About 8885 women were interviewed of which 3962 women were delivered within 5 years before the survey and they were interviewed for ANC visits [19]. From 3962 women interviewed 3915 were included for analysis of this study while 47 women were excluded due to missed information on the outcome of interest [Fig 1].

### Data collection procedures

The Ethiopia Mini Demographic and Health Survey (EMDHS) was conducted for the second time in Ethiopia in 2019. The DHS Program's standard questionnaires served as the basis for the questionnaires' preparation, which involved modifications to account for Ethiopia's unique population and health issues. The questionnaires were translated into Amarigna, Tigrigna, and Afaan Oromo after they were finalized in English [19].

Basic demographic data, such as age, sex, education level, and relationship to the head of the household, were gathered for each individual on the list. The Household Questionnaire's

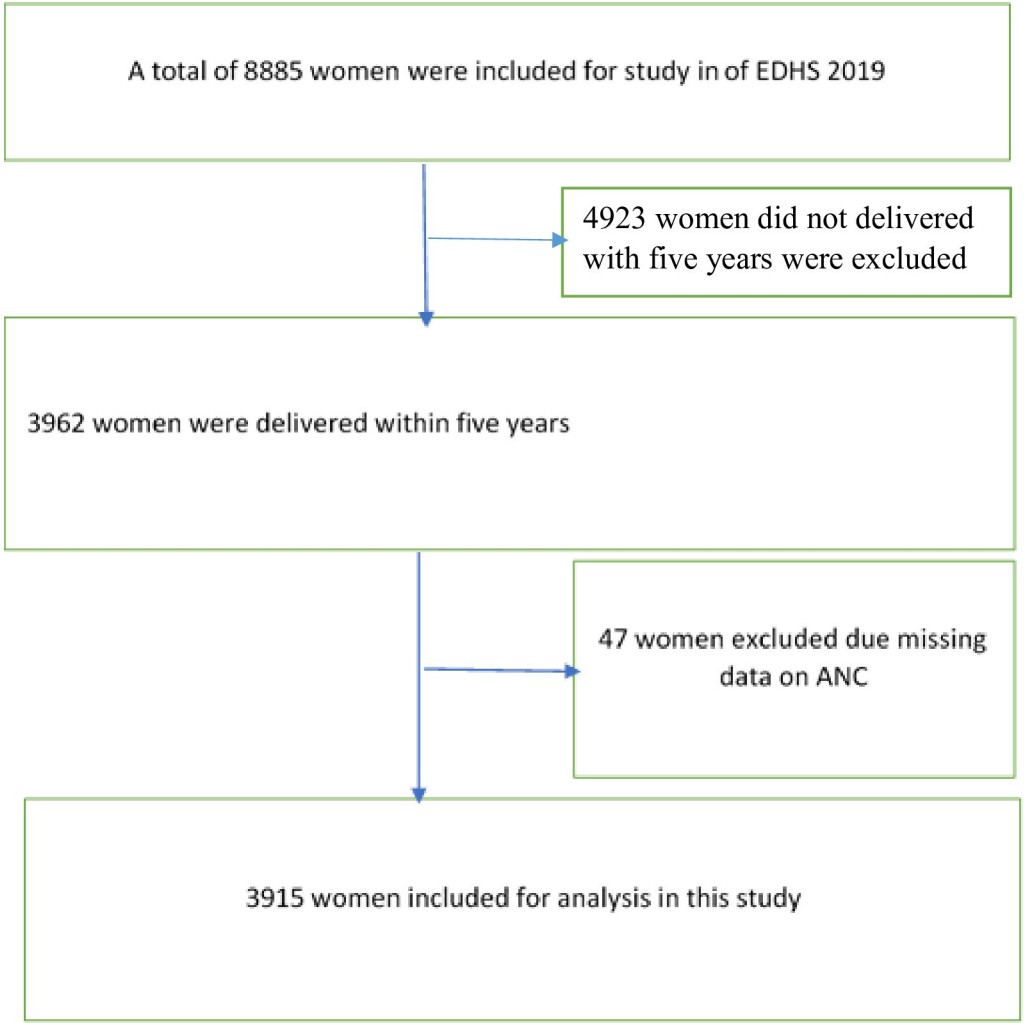

**Fig 1. The process of participant inclusion and exclusion from EDHS 2019.**

data on the gender and age of household members was utilized to determine which women qualified for individual interviews. Information on the features of the household's dwelling unit, such as the type of toilet facilities, the materials used for the floor, and the possession of different durable goods, was also gathered using the Household Questionnaire [19].

The Woman's Questionnaire was used to collect information from all eligible women aged 15–49. The primary subjects of inquiry for these women were: background details; reproduction; contraception; pregnancy, antenatal care, and postpartum care. Responses were recorded during the 2019 EMDHS interviews using tablet computers by the interviewers. The tablets had Bluetooth capabilities, which allowed for remote electronic file transfers between interviewers and supervisors and between interviewers and completed questionnaires within the computer-assisted personal interviewing (CAPI) system. The DHS Program used the mobile version of the Census and Survey Processing (CSPro) System to develop the electronic data collection system used in the 2019 EMDHS. The DHS Program, CSPro, and the US Census Bureau collaborated to develop the CSPro software [19].

## Data quality assurance

The 2019 EMDHS trainers' course took place in Adama from February 11–20, 2019. It consisted of fieldwork, and paper-and CAPI-based in-class training. The fieldwork was carried out in Adama within the clusters that were excluded from the EMDHS sample for 2019. The training of trainers was attended by a total of seventeen trainees. All of the trainees had some background in conducting household surveys, either from participation in earlier DHS surveys conducted in Ethiopia or from surveys using comparable protocols. Lessons learned from the exercise were integrated into the questionnaires for the main training after a debriefing with the trainee field staff took place after field practice [19].

The EMDHS main training was conducted from February 27 to March 19, 2019, at the Central Hotel in Hawassa. For the primary fieldwork, EPHI hired and trained 151 health professionals to work as field supervisors, regional coordinators, female interviewers, female CAPI supervisors, and field supervisors, among other roles. The training's main goal was to equip participants with the skills necessary to administer questionnaires based on paper and CAPI [19].

In-class mock interviews, practice interviews with actual respondents in areas outside the survey sample, a thorough review of the questionnaire content, instructions on how to administer the paper and CAPI questionnaires, and instructions on interviewing techniques and field procedures made up the training course. Additionally, training was provided in fieldwork coordination and data quality control procedures for regional coordinators, field supervisors, and CAPI supervisors [19].

Data collection for the 2019 EMDHS was done by 25 interviewing teams. One field supervisor, one female CAPI supervisor, two female interviewers, and one female anthropometrist made up each team. In addition to the field teams, 11 regional coordinators were assigned, one for each region. Throughout the fieldwork period, the respective teams were regularly visited by the regional coordinator, who stayed with them to oversee and monitor their work and progress. In addition, ten employees of EPHI oversaw and planned fieldwork tasks. EPHI researchers, an ICF technical specialist, a consultant, and representatives from other organizations, including CSA, FMoH, the World Bank, and USAID, supported the fieldwork monitoring [19].

## Study variables and measurements

The response variable of this study was the number of ANC visits. Both community and individual variables are considered as predictor variables. Community water sources are classified as improved and unimproved water sources, place of residence (urban and rural), and region (the ten Ethiopian regions were recoded into three larger categories agrarian, pastoralist, and metropolis). Four regions namely (Tigray, Amhara, Oromia, and Southern Nations, Nationalities People's Region (SNNPR)) were recoded into the agrarian region. Three regions namely (Afar, Somali, Benishangul, and Gambella) were recorded as pastoralist regions. While, Harari, Addis Ababa, and Dire Dawa are among the metropolises' administration regions.

The individual-level variables: Age, level of education, birth order, wealth index, number of under-five children, family size, place of delivery, mode of delivery, marital status and multiple pregnancies were considered as individual level variable label variables.

## Data analysis

Since EDHS data are hierarchical, using a standard model decreases the standard error of effect size since the sample size is larger [30], which in turn decreases the confidence interval of the estimate which affects significance of the estimate [31, 32]. Women in a cluster may

have similar characteristics to those in the other cluster. This similarity within the cluster will violate the rule of independence of observation and equal variance across the cluster. Therefore, using a multilevel model is best for this data rather than using a standard model which helps to compute both fixed effect and random effect variation simultaneously.

A poison regression model is proposed to use for this study, because of dependent variable is the number of ANC visits, which is a countable variable. However, the assumption of poison regression is not met (data is over-dispersed: variance is larger than mean). Therefore, the negative binomial model is more appropriate than the standard Poisson regression model.

In this analysis four models were fitted to estimate a fixed effect for individual and community-level variables and a random effect for variation among clusters. Since the measure of the variation of the cluster was significant indicating a higher intraclass correlation coefficient (ICC).

A Bayesian multilevel negative binomial regression model was used as it is not dependent on the p-value to determine whether the variable is significant or not. The p-value may lead to imprecise evidence as it depends on small size. The prior information for each coefficient of the variables assumed that normally distributed with zero mean and 10,000 variances.

The data were correlated, having intra-class correlation (ICC) = 23.42% and 25.51% for the null and saturated model, respectively, since the value is greater than 5 percent correlation is significant within clusters [33]. The multilevel fitted with four models: Model I (null model) was done without independent variables. Model II was fitted for variables at the individual level, Model III was fitted for variables at the community level, and Model IV was adjusted for individual and community-level variables. The fourth model was fitted to estimate the independent effects of variables both at the community and individual levels on the number of ANC visits. Adjusted incidence ratio (IRR) with 95% credible intervals (CrI) was used to declare association. The goodness of fit test was assessed using the deviance information criterion (DIC). Variance Inflation Factor (VIF) < 5 was considered to check multicollinearity between the individual and community-level variables. All analysis were using STATA version 17.0 (Stata Corp., College Station, TX, USA).

## Ethics approval and consent to participate

The Ethical clearance was provided by the Ethiopia Health and Nutrition Research Institute Review Board. Informed verbal consent was obtained from each woman. The data for this study was obtained after submitting a protocol explaining the objective of the study to the DHS program online. The details of the ethical issues have been published in the EDHS final report, which can be accessed at: http://www.dhsprogram.com.

## Result

### Characteristics of the study participants

Of the total study subjects, 30.4% of the women were between 25–29 years old, and the mean (SD) age of the participants was 28.7 (.11) years. Concerning their education, more than half of the women do not have formal education (51.3%), while only about 3.8% of them attended higher education. Most of the women were married (93.9%), from agrarian regions (87.5%), and rural areas (73.9%) [Table 1].

Out of the total pregnant women in this study, 26.5% do not have ANC visits during their pregnancy, 21.4% of women have at least four ANC visits, and only 3% have at least eight ANC visits [Fig 2].

**Table 1. The number of antenatal care visits by sociodemographic characteristics of women in Ethiopia from 2019 EDHS.**

| Variable | Categories | Frequency | Percent (%) |
|---|---|---|---|
| Age of women in years | < = 24 | 895 | 22.86 |
| | 25–28 | 1,229 | 31.39 |
| | 29–33 | 818 | 20.89 |
| | >34 | 973 | 24.85 |
| Education level of women | No formal education | 2,049 | 52.34 |
| | Primary | 1,281 | 32.72 |
| | Secondary and above | 585 | 14.94 |
| Marital status of women | Married | 3,630 | 92.72 |
| | Non-married | 285 | 7.28 |
| Region of women's residence | Agrarian | 1,690 | 43.17 |
| | Pastoralist | 1,423 | 36.35 |
| | Metropolis | 802 | 20.49 |
| Residence of women | Urban | 976 | 24.93 |
| | Rural | 2,939 | 75.07 |
| Wealth index of household | Poorest | 1,179 | 30.11 |
| | Poor | 676 | 17.27 |
| | Middle | 577 | 14.74 |
| | Rich | 532 | 13.59 |
| | Richest | 951 | 24.29 |
| Community water source | Improved | 2,646 | 67.59 |
| | Non-Improved | 1,269 | 32.41 |
| Household family size | <5 | 1,903 | 48.61 |
| | > = 5 | 2,012 | 51.39 |
| Birth interval (preceding) | < 24 months | 718 | 23.33 |
| | 24–36 months | 863 | 28.05 |
| | > 36 months | 1,496 | 48.62 |
| Place of delivery | Home | 1,780 | 45.47 |
| | Facility | 2,135 | 54.53 |
| Mode of delivery | C/S | 3,638 | 92.92 |
| | SVD | 277 | 7.08 |
| Multiple pregnancies | No | 3,851 | 98.37 |
| | Yes | 64 | 1.63 |

C/S- cesarian section, SVD -spontaneous vaginal deliver

## Factors associated with the number of ANC visits in Ethiopia

**Individual level determinants.** The number of ANC visits was identified to be higher among women in the age group of 25–28 years, 29–33 years, and ≥34 years than women in the group ≤24 years by 13.2% (IRR: 1.13; 95% CrI: 1.11, 1.16), (IRR: 1.15; 95% CrI: 1.15, 1.16) and (IRR:1.14: 95% CrI: 1.12, 1.17), respectively. Women who have attended primary education were shown to have an increased number of ANC visits by 22% (IRR: 1.22, 95% CrI: 1.22, 1.23) compared with those who did not have attended formal education, while attending secondary school and above shown to increase the number of ANC visit by 26% (IRR: 1.26, 95% CrI: 1.26, 1.26) [Table 2].

Women who delivered in a health facility were two times more likely to have a higher number of ANC visits compared with women who delivered at home (IRR: 1.93; 95% CrI: 1.92,

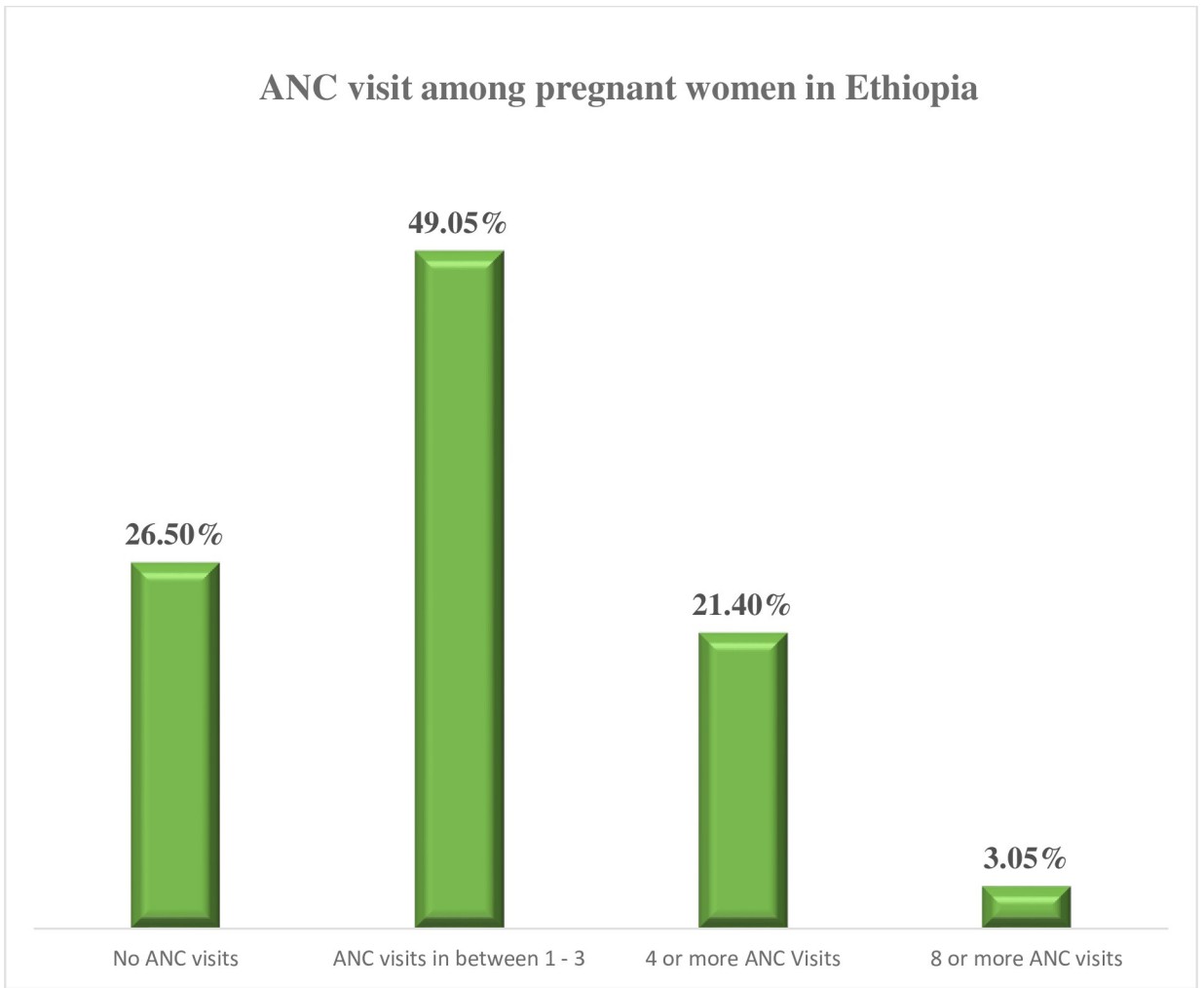

**Fig 2. Number of ANC visit status of women delivered within five years from EDHS 2019.**

1.93). Similarly, women who delivered with cesarian section were shown to have a higher number of ANC visits compared with women delivered with spontaneous vaginal delivery by 18% (IRR: 1.18; 95% CrI: 1.18, 1.19). Women with multiple (twin) pregnancies were shown to have a higher number of ANC visits by 11% (IRR: 1.11; 95% CrI: 1.10, 1.12) compared with women who have singleton pregnancies. Concerning birth interval of preceding pregnancy women with an interval greater than 36 months were shown to have a higher number of ANC visits by 22% (IRR: 1.22; 95% CI: 1.21, 1.23) compared with women having birth interval of less than 24 months [Table 2].

Women from households who have a total family size of $\geq 5$ were shown to have decreased the number of ANC visits by 9.4% (IRR: 0.92; 95% CrI: 0.91, 0.92) compared with women from households who have a total family size $< 5$. Concerning the wealth index, it is revealed that women from the richest, rich, middle, and poor households were shown to have increased number of ANC visits by 28.8% (IRR:1.23; 95% CrI: 1.23, 1.24), 29.4% (IRR: 1.34; 95% CrI: 1.30, 1.37), 34.2% (IRR:1.29; 95% CrI: 1.28, 1.31) and 24% (IRR: 1.29, 95% CI:1.29, 1.30), respectively compared to women from poorest households [Table 2].

**Table 2. Multivariable analysis of factors associated with the number of ANC visits from 2019 mini-demographic data in Ethiopia.**

| Variable | Categories | ModeI I (null) | Model II aIRR (CI) | Model III aIRR (CI) | Model IV aIRR (CI) |
|---|---|---|---|---|---|
| Age of women in years | < = 24 | | 1 | | 1 |
| | 25–28 | | 1.194 (1.185, 1.203) * | | 1.132 (1.112, 1.156) * |
| | 29–33 | | 1.226 (1.219, 1.235) * | | 1.152 (1.147, 1.156) * |
| | >34 | | 1.223 (1.215, 1.230) * | | 1.144 (1.119, 1.174) * |
| Education level of women | No formal education | | 1 | | 1 |
| | Primary | | 1.231 (1.226, 1.238) * | | 1.221 (1.218, 1.222) * |
| | Secondary and above | | 1.240 (1.234, 1.243) * | | 1.261 (1.260, 1.263) * |
| Marital status of women | Married | | 1 | | 1 |
| | Non-married | | 1.057 (1.054, 1.059) * | | 0.999 (0.996, 1.002) |
| Wealth index of household | Poorest | | 1 | | 1 |
| | Poor | | 1.341 (1.332, 1.352) * | | 1.234 (1.227, 1.240) * |
| | Middle | | 1.473 (1.468, 1.480) * | | 1.342 (1.304, 1.370) * |
| | Rich | | 1.455 (1.433, 1.471) * | | 1.294 (1.285, 1.308) * |
| | Richest | | 1.618 (1.614, 1.622) * | | 1.288 (1.286, 1.291) * |
| Household family size | <5 | | 1 | | 1 |
| | ≥5 | | 0.926 (0.918, 0.935) * | | 0.916 (0.915, 0.917) * |
| Birth interval (preceding) | < 24 months | | 1 | | 1 |
| | 24–36 months | | 1.047 (1.047, 1.052) * | | 1.056 (1.051, 1.061) * |
| | > 36 months | | 1.107 (1.103, 1.111) * | | 1.223 (1.211, 1.234) * |
| Place of delivery | Home | | 1 | | 1 |
| | Facility | | 1.853 (1.848, 1.862) * | | 1.931 (1.926, 1.933) * |
| Mode of delivery | C/S | | 1 | | 1 |
| | SVD | | 1.074 (1.064, 1.085) * | | 1.107 (1.100, 1.116) * |
| Multiple pregnancies | No | | 1 | | 1 |
| | Yes | | 1.084 (1.073, 1.093) * | | 1.185 (1.178, 1.195) * |
| Community water source | Improved | | | 1 | 1 |
| | Unimproved | | | 0.736 (0.694, 0.783) * | 0.932 (0.926, 0.940) * |
| Region of women's residence | Agrarian | | | 1 | 1 |
| | Pastoralist | | | 0.709 (0.669, 0.751) * | 0.774 (0.771, 0.779) * |
| | Metropolis | | | 1.021 (0.951, 1.091) * | 1.017 (1.001, 1.036) * |
| Residence of women | Urban | | | 1 | 1 |
| | Rural | | | 0.686 (0.643, 0.733) | 0.920 (0.919, 0.921) * |
| Interclass correlation coefficient (ICC) | | 23.42% | 37.00% | 23.01% | 24.22% |
| Deviance information criteria (DIC) | | 17009 | -614292 | -16442.88 | -614240.7 |

aIRR- adjusted Incidence rate ratio

* p-value less than 0.05, C/S- cesarian section, SVD -spontaneous vaginal delivery

**Community level determinants.** Women from the rural area were shown to have a reduced number of ANC visits by 8% (IRR: 0.92, 95% CI: 0.93, 0.94) compared to women from the urban area [Table 2]. Similarly, women from areas with unimproved water sources were identified to have a reduced number of ANC visits by 6.8% (IRR: 0.92, 95% CI: 0.91, 0.92) compared with women from areas with improved water sources. The number of ANC visits was reduced by 22.6% (IRR: 0.77, 95% CI: 0.77, 0.78) among women from pastoralist regions compared to women from agrarian regions, while it was increased by 1% (IRR:1.01, 95% CI: 1.01, 1.04) among women from metropolis region [Table 2].

## Discussion

In Ethiopia, nearly one-fourth of women did not have ANC visits, and only 3% of women have eight ANC visits during the whole pregnancy, which WHO currently recommends. This finding is slightly higher than the study conducted in 2016 [34]. The difference may be due to variations in the study period, and the recommendation of the new guidelines by WHO in 2016 might have contributed to the difference. A similar data collection tool was used in 2016 and 2019 EHDS but, there is a variation in sample size 3915 in 2019, while it was 7591 in 2016 which might contribute to the difference.

The finding from this study revealed that the number of ANC visits increases with the increasing age of women. The women in the age group of higher than 24 years were shown to have a higher count of ANC visits than women in the age group of ≤ 24 years. This finding is supported by a study conducted in Nigeria and Malawi in 2017 [35]. At the same time, it is contradicted by another study conducted in Ethiopia, which revealed that the likelihood of ANC visits decreases among women in the age group of 35 to 49 years and increases among women between the ages of 15 to 19 years [34]. The difference might be due to the different models used for this data, and the authors used the Bayesian model, which is a more appropriate and representative sample simulated.

This study also identified that the number of ANC visits improves with women's educational level, and women who have attended formal education were shown to have an increased number of ANC visits compared with women who did not. Previous studies in Ethiopia also support the current finding [4, 36, 37]. This finding is also supported by a study conducted in sub-Saharan Africa from 2008 to 2019 [36, 38]. Further, it is consistent with a study conducted in Bangladesh in 2020 [10] and Nigeria in 2020 [39]. Similarly, another study conducted in Indonesia also identified that educated women utilize more ANC visits than women who did not have formal education [40]. This might be because educated women have a higher chance of receiving information on the benefits of ANC visits because of their exposure to mass media [38]. In addition, women who have formal education may know the benefit of ANC visits to the health of mothers and their newborns.

ANC visits increase as the family wealth index increases from poor to rich. This study revealed that women from rich, middle-class families have a higher number of ANC compared with those from poor families. The finding aligns with previous studies conducted in Ethiopia [4, 37, 41]. Similarly, a population-based study conducted in Guinea also reported that educated women are more likely to utilize ANC than non-educated women [42]. Further, this finding is supported by a study conducted in East African countries [43] and a population-based study from Nepal [39]. This might be because women from rich wealth index were more likely to be exposed to mass media, and in addition to this, women from rich families are more likely to have access to health facilities [44].

Women from rural areas were shown to have a reduced number of ANC visits compared with urban residents. This finding is supported by different studies conducted in Ethiopia [27, 37, 41], Guinea [45], Pakistan [46], the Philippines, and Indonesia [47]. This may be because women from rural areas have low access to health facilities and inadequate information on the importance of having an adequate number of ANC visits.

The number of ANC visits was reduced among women from the pastoralist region compared to women from the agrarian region, while it was increased among women from the metropolis region. Even though different studies conducted in Ethiopia have not reported using the current classification of regions, three studies reported that being in the dominating rural regions was less likely to have an increased number of ANC visits than the metropolis

region [6, 19, 41]. This may be because women in the pastoralist region are less likely to attend education compared with agrarian and metropolis regions.

### Limitation of the study

Recall bias may be present because of the cross-sectional design of the study, which makes it impossible to determine the precise cause-and-effect relationship between the number of ANC and its predictors. The study's inability to evaluate certain significant variables, such as the distance to the health facility, the husband's educational status, and media exposure.

### Conclusion

In this study, out of the total ANC candidate women, about 26.5% did not have ANC visits during their pregnancy, and only 3% were shown to have eight and above ANC visits. Factors like, the ages of the women 25–28, 29–33, and ≥34 years, being a primary school, secondary school, and above, delivered in a health facility, delivered with caesarian section, multiple pregnancies, rich, middle and poor wealth indexincome family, were significantly associated with the higher number of ANC visits, while households with large family size and rural residence were significantly associated with a lower number of ANC visits in Ethiopia. Therefore, improving women's education, empowering household income, and providing health education, especially for women in pastoralist regions, may increase the number of ANC visits recommended by WHO; thereby, maternal and child health will be improved.

### Author Contributions

**Conceptualization:** Daniel Atlaw, Tesfaye Getachew Charkos.

**Data curation:** Daniel Atlaw, Tesfaye Getachew Charkos, Jeylan Kasim, Vijay Kumar Chatu.

**Formal analysis:** Daniel Atlaw, Tesfaye Getachew Charkos, Jeylan Kasim, Vijay Kumar Chatu.

**Funding acquisition:** Daniel Atlaw.

**Investigation:** Daniel Atlaw, Vijay Kumar Chatu.

**Methodology:** Daniel Atlaw, Tesfaye Getachew Charkos, Vijay Kumar Chatu.

**Project administration:** Daniel Atlaw.

**Software:** Daniel Atlaw.

**Supervision:** Daniel Atlaw, Tesfaye Getachew Charkos, Jeylan Kasim, Vijay Kumar Chatu.

**Validation:** Daniel Atlaw, Tesfaye Getachew Charkos, Jeylan Kasim.

**Visualization:** Daniel Atlaw, Tesfaye Getachew Charkos.

**Writing – original draft:** Daniel Atlaw.

**Writing – review & editing:** Daniel Atlaw, Tesfaye Getachew Charkos, Jeylan Kasim, Vijay Kumar Chatu.

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
