## [Decision Letter · Decision Letter 0]

26 Dec 2023

PONE-D-23-38385Why does the number of antenatal care visits in Ethiopia remain low? A Bayesian multilevel approachPLOS ONE

Dear Dr. Atlaw,

Thank you for submitting your manuscript to PLOS ONE. After careful consideration, we feel that it has merit but does not fully meet PLOS ONE’s publication criteria as it currently stands. Therefore, we invite you to submit a revised version of the manuscript that addresses the points raised during the review process.

We look forward to receiving your revised manuscript.

Kind regards,

Kahsu Gebrekidan

Academic Editor

PLOS ONE

Journal Requirements:

Additional Editor Comments:

**Please address the comments from both reviewers, find comments from one of the reviewers is on the attached document.**

Reviewers' comments:

Reviewer's Responses to Questions

**Comments to the Author**

1. Is the manuscript technically sound, and do the data support the conclusions?

Reviewer #1: Yes

Reviewer #2: Yes

2. Has the statistical analysis been performed appropriately and rigorously? 

Reviewer #1: Yes

Reviewer #2: Yes

3. Have the authors made all data underlying the findings in their manuscript fully available?

Reviewer #1: Yes

Reviewer #2: Yes

4. Is the manuscript presented in an intelligible fashion and written in standard English?

Reviewer #1: Yes

Reviewer #2: Yes

5. Review Comments to the Author

Reviewer #1: Thank you for providing this opportunity for me to review this manuscript. Please see my comments as follows:

Abstract

1. I would like to see the better interpretation of results, "significant relationship with ANC", please mention lower or better ANC.

Introduction

1. Please indicate the insufficient ANC visits is due to lack of access or lack of information about important of ANC.

2. Please provide some information about the current situation ANC in Ethiopia, for instance a midwife has responsibility for how many pregnant women and if there is a sufficient public health care for ANC.

3. The gap of study is not clear. Previous study showed that 32% of pregnant women did not receive ANC, so what problem authors want to resolve by conducting this study?

Methods

1. The data collection conducted around 4 years ago, that means many things have changed so far. Why authors did not publish or even up-date the data collection?

2. Authors mentioned that they recruited women with age range of 15-45, and then n line 130, they classified women into four groups of <24, 25-28, 29-33 and >34. Did not they have any woman under 18?

3. Please provide more information about method of data collection, interview and so on. Who conducted the interviews? What was instrument, how authors checked the validity of the questionnaire?

4. Did authors used any other documents rather than interview with participants?

Results

1. Where is table 1?

2. Table 2 shows that the rate of CS is so high, please provide reason for this matter?

3. As we know the age of level of education of husband is an important factor for ANC. Please provide those information in the results section.

Discussion

1. What were the limitations of the study?

2. I would like to see the comparison between results of 2019 and 2016 in terms of sample size and instruments.

Reviewer #2: Author has demonstrated a significant finding. A large survey in the EDHS will give a lot of findings if we can analyze the data properly. However, it still found some parts to be improved. Please find attached of our comments in the pdf attachment.

6. PLOS authors have the option to publish the peer review history of their article (what does this mean?). If published, this will include your full peer review and any attached files.

Reviewer #1: **Yes: **Parvin Abedi

Reviewer #2: No

---

## [Author Response · Author response to Decision Letter 0]

15 Jan 2024

Response to reviewers and editors 

Dear respected editors and reviewers 

I would like to say thank you very much on the behalf of all authors of this manuscript, your detailed comments and suggestions has significantly improved our manuscript, if there is still any concern we are knee to hear from you. 

Additional Editor Comments:

Please address the comments from both reviewers, find comments from one of the reviewers is on the attached document.

Response: We have edited the document as suggested by reviewers 

Reviewers' comments:

Reviewer's Responses to Questions

Comments to the Author

1. Is the manuscript technically sound, and do the data support the conclusions?

Reviewer #1: Yes

Reviewer #2: Yes

2. Has the statistical analysis been performed appropriately and rigorously?

Reviewer #1: Yes

Reviewer #2: Yes

3. Have the authors made all data underlying the findings in their manuscript fully available?

Reviewer #1: Yes

Reviewer #2: Yes

4. Is the manuscript presented in an intelligible fashion and written in Standard English?

Reviewer #1: Yes

Reviewer #2: Yes

5. Review Comments to the Author

Reviewer #1: Thank you for providing this opportunity for me to review this manuscript. Please see my comments as follows:

Dear respected reviewer1, thank you very much for your comments that helped us to improve the manuscript.

Abstract

1. I would like to see the better interpretation of results, "significant relationship with ANC", please mention lower or better ANC. 

Response: Thank you very much for your comment, here we amended the result and conclusion section accordingly “. In this study the age of the women 25-28, 29-33, and ≥34 years, being a primary school, secondary school and above, delivered in health facility, delivered with caesarian section, multiple pregnancy, rich, middle and poor income family, were significantly associated with the higher number of ANC visits, while households with large family size and rural residence were significantly associated with lower number of ANC visits in Ethiopia.”

Introduction

1. Please indicate the insufficient ANC visits is due to lack of access or lack of information about important of ANC.

Response: Thank you very much for your comment, 

Some studies identified factors like lack of information on importance of ANC, distance from health facilities, and culture as main contributors for low coverage.

2. Please provide some information about the current situation ANC in Ethiopia, for instance a midwife has responsibility for how many pregnant women and if there is a sufficient public health care for ANC.

Response: Thank you very much for your comment, 

Currently, in Ethiopia the overall coverage of ANC visit is nearly 74%, but number of four ANC visit is around 43%. In terms of access to ANC service, the ratio of health facility including health to pregnant women is 1:185, while number of midwife to pregnant women is 1:162.

3. The gap of study is not clear. Previous study showed that 32% of pregnant women did not receive ANC, so what problem authors want to resolve by conducting this study?

Response: Thank you very much for your comment, here we amended the introduction section as stated below. Currently the issue of ANC is not only the coverage of ANC, number of ANC visits is becoming more important “the previous studies were conducted with older recommendation of WHO, which emphasis on focused four ANC visits. Currently WHO has revised the previous guideline with intention to increase number of ANC visits to eight and more.” That is why the authors focused on number of ANC visits than overall coverage of ANC visits.

Methods

1. The data collection conducted around 4 years ago, that means many things have changed so far. Why authors did not publish or even up-date the data collection?

Response: Thank you very much for your comment, 

 The author have not collected the data by themselves it is taken from Demographic health survey only after requesting data for analysis. This data is national data which is collected at national level authors used the data further analysis and method of data collection and questioner is available online under demographic health survey.

2. Authors mentioned that they recruited women with age range of 15-45, and then n line 130, they classified women into four groups of <24, 25-28, 29-33 and >34. Did not they have any woman under 18? 

Response: Thank very much for your comment

The age is classified based on quartile as explained in the method section since data is collected as continuous data we classified based on quartile and it is recommended to use quartile for grouping continuous data, and fortunately for this data we do not have under 18.

3. Please provide more information about method of data collection, interview and so on. Who conducted the interviews? What was instrument, how authors checked the validity of the questionnaire?

Response: Thank very much for your comment

The data collection is validated tool which is used international to collect demographic health survey as we have explained earlier the data is not collected by authors, it is collected by central statistics agency of Ethiopia, and we have added more detail of data collection process.

4. Did authors used any other documents rather than interview with participants?

Response: Thank very much for your comment

No the detail of data collection process was added in the main manuscript as data collection procedure and data quality assurance. 

Results

1. Where is table 1?

Response: Thank you very much for your comment

It is corrected, it was editorial problem table 1 was written as table 2

2. Table 2 shows that the rate of CS is so high, please provide reason for this matter?

Response: Thank you very much for your comment, it is corrected

This also problem while copy editing from Stata output to word document, the value of SVD was written for C/S

3. As we know the age of level of education of husband is an important factor for ANC. Please provide those information in the results section.

Response: Thank you very much for your comment, yes it very important variable as you said but in this data the variable was not collected, and as we have explained earlier this data mini dhs data in which only some of the variable will be collected this makes it different from the DHS data of 2016

Discussion 

1. What were the limitations of the study?

Response: Thank very much for your comment

“Limitation of the study: Recall bias may be present because of the cross-sectional design of the study, which makes it impossible to determine the precise cause-and-effect relationship between number of ANC and its predictors. The study's inability to evaluate certain significant variables, such as the distance to the health facility, husband’s educational status and media exposure. Further, it is long time since 2019 data was collected, therefore the information provided may not represent the current situation of ANC in Ethiopia. “

2. I would like to see the comparison between results of 2019 and 2016 in terms of sample size and instruments.

Response: Thank very much for your comment

“Similar data collection tool is used in 2016 and 2019 EHDS but, there is variation in sample size 3915 in 2019, while it was 7591 in 2016 that might contribute the difference.”

Reviewer #2: Author has demonstrated a significant finding. A large survey in the EDHS will give a lot of findings if we can analyze the data properly. However, it still found some parts to be improved. Please find attached of our comments in the pdf attachment.

Dear respected reviewer2, thank you very much for your comments that helped us to improve the manuscript.

1. Why does the number of antenatal care visits in Ethiopia remain low? : A Bayesian multilevel approach. This title may be is more suitable for this article

Response: Thank very much for your comment, we have revised as suggested

2. Author can give an underline how much this less frequent is happened in Ethiopia?

Response: Thank very much for your comment, we have mentioned as follows” However, nearly 57% of women received less than four ANC visits in Ethiopia”

3. Give the number of Mean (SD)

Response: Thank very much for your comment, we have added the SD the mean age with SD 28.7 (.11) years. 

4. Is this result being collected and may represent the country ANC rate?

Response: Thank very much for your comment, yes the data is collected it representative. 

5. What does it mean for these all results? Author can add one additional sentence to explain with

Response: Thank very much for your comment, we have added phrases to explain more the result section of the abstract.

We have added the following sentence in the conclusion section of abstract “This result is much lower as compared to WHO recommendation, which states that all pregnant women should have at least eight ANC visits”

6. This conclusion is shown differently with the results

Response: Thank very much for your comment, here we have revised as follows

“In this study the age of the women 25-28, 29-33, and ≥34 years, being a primary school, secondary school and above, delivered in health facility, delivered with caesarian section, multiple pregnancy, rich, middle and poor income family were significantly associated with the higher number of ANC visits, while households with large family size and rural residence were significantly associated with lower number of ANC visits in Ethiopia.”

7. This writing style should be consistent whether a space before citation number or not. Check all spelling in the whole article.

Response: Thank very much for your comment, we have revised the whole document for punctuation and spelling, proof read was done repeatedly.

8. Give the readers the updated findings about the main problem of ANC rate in Ethiopia.

Response: Thank very much for your comment, here we have added a paragraph explaining main problem of ANC

“Currently, in Ethiopia the overall coverage of ANC visit is nearly 74%, but number four ANC visit is around 43%.Currentely, in Ethiopia the overall coverage of ANC visit is nearly 74%, but number four ANC visit is around 43%. In terms of access to health facilities, the ratio of health facility including health to pregnant women is 1:185, while number of midwife to pregnant women is 1:162. Some studies identified factors like lack of information on importance of ANC, distance from health facilities as main contributors for low coverage”

9. Author can explain more detail in how much increase it would be

Response: Thank very much for your comment

10. Author can also add the percentage in Parentheses.

Response: Thank very much for your comment, we have added percentage in parenthesis 

11. How about the data collection during EDHS? May be a summary about this information can be added.

Response: Thank very much for your comment, here as follows we have added detail of data collection process and data quality assurance 

“Data collection procedures 

The Ethiopia Mini Demographic and Health Survey (EMDHS) is conducted for the second time in Ethiopia in 2019. The DHS Program's standard questionnaires served as the basis for the questionnaires' preparation, which involved modifications to account for Ethiopia's unique population and health issues. The questionnaires were translated into Amarigna, Tigrigna, and Afaan Oromo after they were finalized in English.

Basic demographic data, such as age, sex, education level, and relationship to the head of the household, were gathered for each individual on the list. The Household Questionnaire's data on gender and age of household members was utilized to determine which women qualified for individual interviews. Information on the features of the household's dwelling unit, such as the type of toilet facilities, the materials used for the floor, and the possession of different durable goods, was also gathered using the Household Questionnaire.

The Woman’s Questionnaire was used to collect information from all eligible women age 15-49. The primary subjects of inquiry for these women were: background details; reproduction; contraception; pregnancy, antenatal care and postpartum care. Responses were recorded during the 2019 EMDHS interviews using tablet computers by the interviewers. The tablets had Bluetooth capabilities, which allowed for remote electronic file transfers between interviewers and supervisors and between interviewers and completed questionnaires within the computer-assisted personal interviewing (CAPI) system. The DHS Program used the mobile version of the Census and Survey Processing (CSPro) System to develop the electronic data collection system used in the 2019 EMDHS. The DHS Program, CSPro, and the US Census Bureau collaborated to develop the CSPro software.

Data quality assurance 

The 2019 EMDHS trainers' course took place in Adama from February 11–20, 2019. It consisted of field work, and paper-and CAPI-based in-class training. The fieldwork was carried out in Adama within the clusters that were excluded from the EMDHS sample for 2019. The training of trainers was attended by a total of seventeen trainees. All of the trainees had some background conducting household surveys, either from participation in earlier DHS surveys conducted in Ethiopia or from surveys using comparable protocols. Lessons learned from the exercise were integrated into the questionnaires for the main training after a debriefing with the trainee field staff took place after field practice.

The EMDHS main training was conducted from February 27 to March 19, 2019, at Central Hotel in Hawassa. For the primary fieldwork, EPHI hired and trained 151 health professionals to work as field supervisors, regional coordinators, female interviewers, female CAPI supervisors, and field supervisors, among other roles. The training's main goal was to equip participants with the skills necessary to administer questionnaires based on paper and CAPI. 

In-class mock interviews, practice interviews with actual respondents in areas outside the survey sample, a thorough review of the questionnaire content, instructions on how to administer the paper and CAPI questionnaires, and instructions on interviewing techniques and field procedures made up the training course. Over the course of four days, teams completed the anthropometry component of CAPI field practice. Additionally, training was provided in fieldwork coordination and data quality control procedures for regional coordinators, field supervisors, and CAPI supervisors 

Data collection for the 2019 EMDHS was done by 25 interviewing teams. One field supervisor, one female CAPI supervisor, and two female interviewers. In addition to the field teams, 11 regional coordinators were assigned, one for ea

---

## [Decision Letter · Decision Letter 1]

28 Feb 2024

PONE-D-23-38385R1Why does the number of antenatal care visits in Ethiopia remain low?: A Bayesian multilevel approachPLOS ONE

Dear Dr. Atlaw,

Thank you for submitting your manuscript to PLOS ONE. After careful consideration, we feel that it has merit but does not fully meet PLOS ONE’s publication criteria as it currently stands. Therefore, we invite you to submit a revised version of the manuscript that addresses the points raised during the review process.

We look forward to receiving your revised manuscript.

Kind regards,

Kahsu Gebrekidan

Academic Editor

PLOS ONE

Additional Editor Comments:

The decision at this level is major revision again because one of the pervious reviewers could not agree to review the revised version, so that we had to invite another reviewer.

Reviewers' comments:

Reviewer's Responses to Questions

**Comments to the Author**

1. If the authors have adequately addressed your comments raised in a previous round of review and you feel that this manuscript is now acceptable for publication, you may indicate that here to bypass the “Comments to the Author” section, enter your conflict of interest statement in the “Confidential to Editor” section, and submit your "Accept" recommendation.

Reviewer #2: All comments have been addressed

Reviewer #3: (No Response)

2. Is the manuscript technically sound, and do the data support the conclusions?

Reviewer #2: Yes

Reviewer #3: Yes

3. Has the statistical analysis been performed appropriately and rigorously? 

Reviewer #2: Yes

Reviewer #3: Yes

4. Have the authors made all data underlying the findings in their manuscript fully available?

Reviewer #2: Yes

Reviewer #3: Yes

5. Is the manuscript presented in an intelligible fashion and written in standard English?

Reviewer #2: Yes

Reviewer #3: No

6. Review Comments to the Author

Reviewer #2: This revision manuscript have addressed all the reviewer comment. This version can go through to the next stage.

Reviewer #3: Dear colleagues,

Thank you for this paper on ANC visits in Ethiopia using Bayesian multilevel approach. While the paper posits exciting findings, the introduction, methodology and discussion sections needs improvement and proofreading. Please refer to the track changes and comments. Some parts of the introduction was written with inaccurate facts and ambiguous statements, which made summarizing the gap in research on ANC visits poorly described. While the quantitative methods of the study are outlined, imprecise and inconcise writing renders the section hard to read. It would be best for the paper to have a tighter rewrite and undergo another round of review before being considered for publication.

7. PLOS authors have the option to publish the peer review history of their article (what does this mean?). If published, this will include your full peer review and any attached files.

Reviewer #2: No

Reviewer #3: No

---

## [Author Response · Author response to Decision Letter 1]

4 Mar 2024

Response to reviewer 

Dear respected reviewers and editors of PLOS one 

Thank you very much for your detailed and genuine comments and suggestions 

Point by point response to reviewer # 3

1. Surely increasing no of visits is not the panacea to pregnancy complications, as it could instead encourage the overuse of antenatal service. Response: Thank you very much for your comment, yes increasing number of ANC visits alone is not a solution for reducing pregnancy complication but as number visits increase the service given for pregnant women is of various type and time of visits also mater the service to be delivered.

2. It could not possibly be 8, check your sources. Response: Thank you very much for your comment, this is new guideline of WHO developed in 2016 that suggests/recommend minimum visits to be 8/ the previous guideline which suggests minimum 4 visits were modified 

3. This is a strange statement. It is highly unusual even in developed countries to have a pregnant mother making 8 or more ANC visits (unless she has complications requiring so). Thus the inclusion of this percentage should not be made if it is meant to point out that the rate of minimum ANC visits (which is 4, not 8 btw) is inadequate in Sub-Saharan Africa. Response: Thank you very much for your comment, this new recommendation that is why we have conducted this study (https://www.who.int/news/item/07-11-2016-new-guidelines-on-antenatal-care-for-a-positive-pregnancy-experience )

4. This sentence is confusing. Suggest to include a sentence highlighting the research gap, elaborating on the previous sentence about women’s lack of access to health services and low ANC visits. E.g Previous studies have primarily focused on the utilization level, timing of initiation, and associated factors of ANC visits, rather than specifically examining the determinants of the number of visits. This study aims to fill this gap by identifying both community and individual-level determinants influencing the number of ANC visits in Ethiopia. Response: Thank you very much for your comment here is it modified as you stated “Different studies were conducted in Ethiopia, but the previous studies were conducted on utilization level, time of initiation and associated factors rather than focusing on the number of visits, which may result in loss of information in classification count data. Further, the previous studies were conducted with the older recommendation of WHO, which emphasis on foncused four ANC visits. Currently, WHO has revised the previous guideline to increase the number of ANC visits to eight or more (28). Therefore, this study, aimed to identify community and individual-level determinants of the number of ANC visits in Ethiopia.” And the mentioned sentence was improved as follows “Some studies identified factors like lack of information on the importance of ANC, distance from health facilities, and culture as main contributors to low coverage”

5. Study or studies? There is only one citation here Response: Thank you very much for your comment it is corrected 

6. To improve in what way? I’m concerned with the implication linking the two variables (ANC visits & MNH improvements). Best to be nuanced when interpreting the correlations. Correlation may not imply causation

Response: Thank you very much for your comment it is corrected as follows “as the number of ANC visits increases, maternal and neonatal morbidity and mortality health were shown to decrease”

7. These descriptive variables read like a laundry list. Consider simplifying the paragraph or presenting in in a table. 

Response: Thank you very much for your comment it is simplified

8. Affect in what way?

Response: Thank you very much for your comment it is simplified, as ICC increases it increases the standard error which intern increases the confidence interval their by as confidence interval increases the estimate may become significant.

9. This part needs to be more concise, as model II was written twice. Eg. There were four multilevel models fitted in this study on the number of ANC visits: non-independent null model (Model I), individual-level variables model (Model II), community-level variables model (Model III), and combined individual and community-level variables (Model IV).

Response: Thank you very much for your comment it is corrected.

---

## [Decision Letter · Decision Letter 2]

9 Apr 2024

Why does the number of antenatal care visits in Ethiopia remain low?: A Bayesian multilevel approach

PONE-D-23-38385R2

Dear Mr Daniel,

We’re pleased to inform you that your manuscript has been judged scientifically suitable for publication and will be formally accepted for publication once it meets all outstanding technical requirements.

Kind regards,

Kahsu Gebrekidan

Academic Editor

PLOS ONE

Additional Editor Comments (optional):

Reviewers' comments:

Reviewer's Responses to Questions

**Comments to the Author**

1. If the authors have adequately addressed your comments raised in a previous round of review and you feel that this manuscript is now acceptable for publication, you may indicate that here to bypass the “Comments to the Author” section, enter your conflict of interest statement in the “Confidential to Editor” section, and submit your "Accept" recommendation.

Reviewer #3: All comments have been addressed

2. Is the manuscript technically sound, and do the data support the conclusions?

Reviewer #3: Yes

3. Has the statistical analysis been performed appropriately and rigorously? 

Reviewer #3: Yes

4. Have the authors made all data underlying the findings in their manuscript fully available?

Reviewer #3: Yes

5. Is the manuscript presented in an intelligible fashion and written in standard English?

Reviewer #3: Yes

6. Review Comments to the Author

Reviewer #3: The latest revision has addressed all the concerns raised in the previous draft. The work presented meets the requirements for publication.

7. PLOS authors have the option to publish the peer review history of their article (what does this mean?). If published, this will include your full peer review and any attached files.

Reviewer #3: No
